# Social Participation and Loneliness in Older Adults in a Rural Australian Context: Individual and Organizational Perspectives

**DOI:** 10.3390/ijerph21070886

**Published:** 2024-07-08

**Authors:** Leah Wilson, Carrigan Rice, Sandra Thompson

**Affiliations:** 1Department of Global Health, Georgetown University, Washington, DC 20057, USA; 2Western Australian Centre for Rural Health, University of Western Australia, Geraldton, WA 6009, Australia; sandra.thompson@uwa.edu.au

**Keywords:** social participation, healthy aging, program sustainability, rural stoicism, rural Australia, Aboriginal Australians, rural aging

## Abstract

A rise in aging populations globally calls attention to factors that influence the well-being and health of older adults, including social participation. In Australia, rural older adults face cultural, social, and physical challenges that place them at risk for isolation. Thus, research surrounding social participation and healthy aging is increasingly relevant, especially in rural areas. This qualitative study in a remote town in Western Australia explores barriers and facilitators to older adults’ social participation. To investigate multiple perspectives, 23 adults aged 50+ and 19 organizations from a rural town were interviewed. A stakeholder reference group was engaged to refine the research design and validate the findings. Feedback from early interviews was used to refine the data collection process, thus enhancing the validity of the findings. Thematic analysis showed that health and mobility issues, inadequate infrastructure, poor sustainability, and cultural tensions commonly impacted social participation. Themes of rural town culture, cultural power dynamics, and rural stoicism were identified as cultural aspects that inhibited participation. Based on results of this study and the supporting literature, recommendations for inclusive activities include supporting community-designed programs, utilizing culturally sensitive language and personnel, expanding services using existing community resources, and diversifying older adults’ roles in existing groups.

## 1. Introduction

Awareness of aging and its effects on society has heightened due to global demographic shifts. Worldwide, people live longer than ever before and older people constitute a larger proportion of the population. This aging demographic pattern is observable in Australia, in which 16% of the current population is aged 65 and older [1]. This percentage is expected to increase to over one-quarter of the population by 2066, with exacerbated effects in regional and remote communities [1]. The population change places pressure on social and health services to sustain an increasing demand for medical and aged care for older adults in Australia [1,2]. Thus, research and initiatives to promote the health and well-being of older adults in these regions are essential.

As concerns about aging populations rise globally, national policies have shifted towards promoting healthy aging, including older adults’ social participation. Numerous studies and interventions with a diverse range of methods have been created to address social participation and well-being, targeting societal, community, and individual levels of influence through policies, anti-ageism campaigns, age-friendly communities, transportation services, volunteering interventions, and social groups [3,4,5].

Programs focused on improving social participation can protect mental, cognitive, and physical health among older adults. Social participation can reduce this loneliness and social isolation, which are linked to health and social challenges [6,7,8]. Importantly, it is estimated that 34% of Australian adults feel lonely, making this a widespread issue to address [9]. As a potential support against the challenges of aging, increased participation also helps mitigate stress related to life transitions such as retirement, loss of family members and friends, health problems, and functional decline [10,11]. Additionally, those who participate are more likely to access or be referred to care more quickly for health changes, reducing the number of acute hospital visits for chronic conditions [12,13]. This suggests that programs focusing on social participation may be effective at increasing older adult’s well-being and healthy aging.

Due to geographic isolation and limited access to health and social services, rural older adults likely face additional barriers to health, economic stability, and social participation, such as lack of transport, increased disability, poverty, and limited health services [14]. They often lack social support and experience greater loneliness than those in suburban or metropolitan areas [15,16]. Considering all these factors, older rural adults are at high risk for health issues and increased reliance on already limited services, highlighting the need for initiatives promoting healthy aging and social participation.

Despite these serious challenges for rural adults, research on older adults’ healthy aging and social participation in rural areas is limited, especially in qualitative understanding [10,17]. Due to the gap in research and the geographic isolation of many rural and remote regions of Australia, there is a compelling need for social participation research in these communities. Qualitative research can provide nuanced insight through in-depth conversations with older adults and their care services, which is particularly beneficial in less understood rural populations [10,14]. Thus, this study gained insight into the facilitators and barriers of social participation by capturing participant perspectives on how social participation functions in their town. Our findings contribute to the literature on social participation and healthy aging, informing future initiatives within rural and remote Australia.

## 2. Materials and Methods

### 2.1. Setting

This study took place in a remote town in the Gascoyne region of Western Australia. The coastal town is situated approximately 900 km north of Perth, the state capital; this area is classified as remote and has a population of just under 5000 people, with a high proportion of retirees relative to neighbouring areas. Population estimates show that around 40% of its population is over the age of 50 years and 16% of its population is 65+ [18]. About 16% of the population is identified as Aboriginal, 67% as White, and 17% as unknown [18].

The investigators interviewed both older residents of the town and representatives of local community organizations. Many weeks were spent developing relationships with the local older population and relevant service providers before formal semi-structured in-depth interviews were undertaken throughout October and November 2023.

### 2.2. Recruitment of Older Residents

Older adults were recruited via a local general practice. The clinic was one of two operating medical centres within the town. Each participant, aged 50 and above, was referred by health personnel to speak with researchers to ascertain their interest in participation. This clinic sample was based on patient availability. The discretion of medical providers was used to determine which participants were cognitively able to participate. In addition, any participants who were not English-speaking were excluded.

All potential participants were provided with written information about this study and those who agreed to participate completed written consent forms. During the time of this study, the clinic was conducting health assessments on those aged 75 and older, per Medicare protocol.

### 2.3. Organizational Staff Recruitment

A list of relevant organizations that provide services to older residents of the town was compiled after conducting inquiries with multiple informants, holding discussions with the local Stakeholder Advisory Group, and undertaking a search of relevant websites. Organizational interview recruitment and selection were informed by discussions at a community stakeholder group meeting as well as by consulting the Promising Approaches Loneliness Framework [4]. These approaches ensured the recruitment of a wide range of services that reach older adults within the community. Nineteen of the thirty-nine organizations that were identified and contacted agreed to participate.

Organizations included but were not limited to health services, physical activity groups, arts and creative activity groups, social groups, transportation services, psychological services, governmental programs, mentoring services, social welfare organizations, volunteering organizations, and local businesses.

Organizations and services were initially contacted via email or by phone. Representatives were chosen based on their leadership status within the organization and availability. All follow-up requests for participation were completed in person by investigators. Prior to the interviews, all participants were informed of the purpose of this study, provided with study information forms and consent forms that explained the interview process, and advised that the interview would be recorded. Participants who agreed to participate signed written consent forms.

### 2.4. Data Collection and Analysis

All community member and organizational interviews were conducted in person at either the medical clinic or, in the case of the organizational interviews, at a location convenient to the participants. No personal identifying data were retained for the data analysis. Individual follow-up contact was not made with any older community members who were recruited for this study; however, interviewed organizational representatives received a summary of their transcripts to ensure information was recounted accurately.

All interviews were semi-structured and followed interview guides (Appendix A). Separate interview guides were designed for individual and organizational interviews. Interviews for individuals included the DeJong Gierveld Loneliness Scale (six-item scale), as well as questions related to social participation or engagement in social activities [19]. Questions in the organizational interviews explored barriers and facilitators to participation and followed the Promising Approaches Loneliness Framework [4].

Interviews were recorded, transcribed, and managed by investigators. Field notes from each participant were also recorded to inform analysis.

For analysis, all interviews were categorically and thematically analysed by hand. Two interviewers initially coded transcripts according to the themes within the interview guides, with codes distinct for individual and organizational interviews. Data were analysed using a hybrid inductive–deductive, open coding approach on field notes, annotations, and interview transcriptions [20]. Based on codes found in organizational and community member interviews, overarching themes were identified and further developed. Interviewers then returned to initial transcripts and conducted line-by-line analysis using overarching threads. Based on the interviews, themes were identified concerning structural and cultural barriers that hinder social participation and foment loneliness. Interviewees also identified facilitators and solutions to engagement.

### 2.5. Ethics

All subjects provided informed consent for inclusion before participating in this study. This study was conducted in accordance with the Declaration of Helsinki and the protocol was approved by the University of Western Australia’s Institutional Review Board (ET000810). This study was determined to be low-risk for human subjects.

### 2.6. Reflexivity and Positionality Statement 

The researchers acknowledge educational and racial differences from study participants. In terms of rurality, while the researchers lived in the community during the project, we remained outsiders to this remote area. In addition, the primary authors have age-related privilege and do not have lived experience in aging.

## 3. Results

A total of 38 interviews were conducted, comprising 19 individual and organizational interviews each. These interviews lasted from 8 to 50 min and provided details on the barriers and facilitators to social participation as well as loneliness within the rural town. The overlap in data from these older resident and service provider interviews was strong, thus the themes are presented together, drawing evidence from both individual and organizational perspectives.

For the individual interviews, 28 patients spoke with investigators, and 24 agreed to participate. One participant was later excluded based on location inclusion criteria. Initially, only those who scored a two or above on the loneliness scale were invited to participate in a longer interview. However, as sample sizes were limited, inclusion criteria were expanded to include every questionnaire participant regardless of loneliness score. Thus, we conducted 23 loneliness questionnaires and 19 individual interviews. As displayed in Table 1, most participants were male (70%), over 70 years of age (74%), and non-Aboriginal (96%). Of these community residents, 65% participated in some type of group or activity. During analysis, the authors noted that neither gender, age, nor ethnicity contributed significantly to identified themes.

Table 2 compares the DeJong Gierveld Loneliness Scale scores with four demographic characteristics identified as correlates of loneliness in previous literature [16,20]. In this sample, 39% of participants were classified as lonely. Regarding gender, 43% of women were lonely, compared to 38% of men. The prevalence of loneliness was also more common (41%) in ages 70–99, compared to ages 50–69 (33%). The most notable disparity in loneliness scores was observed in health status: 75% of individuals with poor or fair self-rated health were lonely, whereas only 15% of those in good or very good health screened as lonely. Loneliness status varied across marital status categories, with an even split among those widowed or divorced, while only 29% of married participants experienced loneliness. Finally, 33% of cohabitating participants screened as lonely, while 66% did not.

For organizational interviews, twenty-five social groups and service organizations within the community were contacted, and based on their willingness to participate, 19 organizational representatives completed the interviews. The organizational characteristics of these participants are displayed in Table 3 and Table 4.

Overarching themes from organizational and individual interviews are further outlined below. To contextualize participant quotes, codes used to identify individual interview participants based on age, gender, and loneliness are listed: gender (f, m), age group (50–59, 60–69, 70–79, 80–89, 90–99), and loneliness (L-lonely, NL-not lonely).

### 3.1. Structural Barriers

One central theme was structural barriers that prevented social participation and involvement in activities. Within this theme, both organizations and individuals mentioned sustainability in rural environments as well as within programs, lack of third spaces, and limited infrastructure as key structural barriers.

#### 3.1.1. Sustainability

Participants emphasized the importance of purposeful engagement and volunteering in sustaining clubs and activities. One group leader (m, 80–89, NL) underscored the significance of maintaining attendance for the group’s survival: “I keep going… to make sure we keep up with attendance, otherwise we’d fold”. However, this sense of purpose also evoked distress among some older adults who felt burdened by excessive responsibilities. For instance, a participant (f, 60–69, L) expressed her distress at shouldering too much responsibility due to a lack of younger volunteers: “It’s hard to get anyone to help… My husband is 74 and… [there’s] someone that helps him who is 78. We’ve got no younger people”. In this sense, the reduced supply of available volunteers increased the pressure on existing ones.

Interviewees reported that reliance on itinerant volunteers was difficult and led to the decline in multiple clubs, exemplifying the “revolving door” nature of travellers and “itinerant workers” in rural towns. One participant (m, 70–79, L) recounted the fall of his political group, highlighting a dependency on key volunteers: “The new president, he seems to have got lost–we can’t find him. So we can’t have a meeting without him”. Organization leaders also emphasized a reliance on a core group of volunteers who often juggled multiple roles within various clubs and organizations, which raised concerns about volunteer burnout and overuse. As stated by one organizational participant, “So in that way, it’s the same people… there’ll be like 20 percent doing all the work and everyone else enjoying the benefits”. Participants also highlighted the constant need for volunteers and support to sustain events and activities. Without adequate volunteer support, many programs were unable to run events or operate effectively, which challenged the sustainability of activities. One participant (m, 70–79, L) described the effect of this turnover on his social life: “That’s why it’s hard to grow close to anyone”.

#### 3.1.2. Sustainable Programs

Sustainability emerged as a prominent issue across organizations and programs; a total of 10 of the organizations mentioned at least one problem related to sustainability. One organization leader commented on the transient, high turnover of staff, affecting program continuity: “It is a very transient sort of community. So when staff move, the programs change”. This leader further emphasized that inconsistency erodes community trust and engagement. Describing the effect on engagement, one individual interview participant (f, 70–79, L) compared the exodus of young workers out of the rural town to “rats on a sinking ship. So that’s why I don’t participate in this stuff”.

Further, only six organizations out of the 19 interviewed used existing community assets within their programs, which had consequences for funding and engagement. One organization leader emphasized the importance of grounding organizations in existing resources: “So coming in… and standing on your own and doing things on your own does not work”. Without grounding in the community, programs become unstable, leading to funding gaps and eventual program cessation. One organization leader noted the following: “There are programs… funded for six months… they are not able to continue it and that is where… families are left”. The persistent challenges of transient staffing, poor funding and lack of community grounding result in low engagement with services. Multiple organizational representatives reported cancelled events and poor engagement; for example, one representative stated the following: “In all honesty… I’ve run two [programs] because they got cancelled… the one I had, no one showed up”. As a consequence, failed programs lose funding and perpetuate a lack of community trust in the sustainability and consistency of programs.

#### 3.1.3. Third Spaces

Lack of infrastructure and third spaces, or informal gathering places beyond work or home, emerged as another subcomponent of structural barriers that hindered social interaction and participation. Within organizational and individual interviews, participants noted that they or their clients missed feeling connected to the town and spending time out and about with friends in cafes, shops, or other local businesses. Seven of the organizational interviews highlighted that organizations, group activities, cafes, and other businesses had closed or had limited hours due to economic instability and theft. This led to many “gaps” in informal organic social interactions and social isolation, especially after the hours of formal programming. One participant, a newcomer to the town, emphasized these concerns as follows:

It doesn’t take me that long to get connected to a place but this time it’s been longer. Maybe because there is not a restaurant that you can sit in… with friends… Nothing’s open… there’s no place where you can meet.(f, 80–89, NL)

Other participants mentioned that the lack of accessible third spaces also hindered motivations to socialize. As one service provider stated, “I actually do social support every day, and trying to find a park or cafe near a parking spot, for someone who can’t mobilize… is a big issue. And I think a lot of people go, ‘Oh, well, we’re not going out because we can’t get a park[ing space] close… That would stop a lot of people’”.

These observations suggest that the absence of third spaces was an obstacle to social participation for older people in the community. The desire for increased availability and accessibility of such spaces reflects a broader need for infrastructure, including parks, gardens, and local businesses that support social gatherings and cohesion in rural settings.

### 3.2. Cultural Barriers

A prominent theme within individual and organizational interviews was community and cultural tensions that impacted effective participation within the town. These tensions manifested in many forms, including rural stoicism, town culture, local versus outsider power dynamics, Aboriginal versus non-Aboriginal tensions, and intergenerational trauma and shame.

#### 3.2.1. Town Culture and Rural Stoicism

Individuals grappling with loneliness expressed feelings of social exclusion, reporting a sense of not being fully accepted by the local community of “born and bred” residents (m, 50–59, L). Both participants and organizations characterized the town as “welcoming, not inclusive”, attributing this characteristic to rural Australia: “That’s just how rural towns are. Too bad, so sad” (m, 50–59, L). An organizational leader further elaborated as follows: “People are not so open to just inviting new people into established social circles”.

This social exclusion had consequences relative to the rural context. Due to the limited availability of formal services, participants mentioned relying on friends, family, or fellow group members for informational and functional support, such as driving and technology support. However, those outside established social networks faced difficulties accessing events and services. This was exemplified by one participant’s (f, 70–79, L) struggle to secure transportation for medical treatment due to her self-identified small social network. In this sense, those on the outside of social networks cannot rely on their connections to compensate for the lack of services or formal information in this rural town. This reliance on informal networks was seen as beneficial by some but presented significant challenges for individuals without strong social connections, as highlighted by an organizational interviewee: “I think that’s good if they are socially connected… But it must be very hard for the older people who aren’t”.

Another theme emerging through individual interviews was rural stoicism and associated stigma around help-seeking. When faced with questions regarding feelings, connection, and participation, a sizable number of participants stated that they “don’t need people”, “don’t need assistance”, and they “don’t get lonely” (m/f, 70–79, L; m, 80–89, L). One participant described being especially wary of “needy” people (m, 70–79, L). Despite outward assertions of self-sufficiency, these individuals often scored high on loneliness measures, indicating a disconnect between expressed sentiments and inward emotional experiences.

Resistance to seeking help was often accompanied by negative perceptions of community and social engagement. Some participants with stoic attitudes described the town in disparaging terms, labelling its inhabitants as “stupid, with poor experiences” (m, 70–79, L) or “self-centered gits” (m, 50–59, L), and expressed reluctance to participate in community activities or organizations. Understanding this cultural context of stoicism, organizations tailored their approaches to engage men, acknowledging their reluctance and emphasizing supportive outreach: “Men can be very stubborn, we will… say, ‘Do you need some help?’ and sometimes they’ll say no, but we will… send a referral anyways”. This approach underscores the importance of aligning services with the cultural norms of those benefiting from services, fostering meaningful engagement within the community.

#### 3.2.2. Other Cultural Tensions

Other tensions stemming from in and out groups (urban versus rural and Aboriginal versus non-Aboriginal) were highlighted by organizations to adversely impact program effectiveness. These tensions and power dynamics were further exacerbated by a lack of use of community or cultural knowledge and community-led decision-making. This sentiment was encapsulated by one organization’s observation: “You just come in and have all these great ideas but…nothing happens. It is wasting a lot of money… [and] time and not helping anyone”.

Organizations also emphasized the importance of having staff with local connections to bridge cultural gaps and promote community engagement. One participant from a local government organization made the following comment:

Everyone who works for the [governmental organization] lives in the region, so it doesn’t work if you don’t… Most of the state government… [is] based in Perth, and they fly up and they… work up here, but it doesn’t work. So we [in the organization] know the people… We’re connected to the community… we try and have quite a diverse range of people… so that we’re interfacing with many different parts of the community.

Many Aboriginal organizations also expressed concerns about external research that failed to benefit the community itself or result in meaningful change. It was stated that this caused wariness and reluctance to engage with programs run by external organizations, particularly among Aboriginal elders.

The cultural barriers between the Aboriginal community and the predominantly White town centre were noted to contribute to low program engagement: “When the town goes out there, the Aboriginal community does not engage”. These barriers were further compounded by government programs that overlooked cultural considerations and implemented programs without genuine consultation, leading to a perception of tokenism and deepening community divides. One community member (m, 70–79, L) highlighted this lack of communication: “No one in this town seems to have the guts to say ‘Okay, let’s have a talk’. No one”.

#### 3.2.3. Shame

Additionally, intergenerational trauma and shame associated with historical racism played a significant role in hindering older Aboriginal adults’ participation in predominantly White social settings. An Aboriginal participant made the following comment: “There’s a lot of shame in [Indigenous elders] them to go in, like go to a dancing class or go to a pottery class. Because since colonization, Aboriginal people have been shamed. And it’s just been carried”.

Other organizations also stated that many Aboriginal elders were reluctant to ask for help or admit they are aging due to embarrassment related to help-seeking. This cultural shame was associated with a lack of motivation, poor self-esteem, and poor confidence to leave one’s home and join groups.

We can’t get some people to engage… people are stuck in their ways… it is out of their comfort zone to… join a program that’s not known to them. They would rather be in their homes… or their little area of living.

These cultural barriers emphasize the potential lack of consideration of Aboriginal voices, which hinders social engagement and older adult participation within small-town environments.

### 3.3. What Works: Local Knowledge, Adjusting for Disability, and Empowerment 

During individual and organizational interviews, positive factors that increased social connections and participation emerged as common themes. Common facilitators mentioned included using local knowledge to build programs for older adults, empowering self-motivation, and adjusting activities for changes in functionality.

#### 3.3.1. Local Knowledge and Strong Relationships

One theme brought forward by organizational interviews was the importance of using strong relationships and local knowledge. Sixteen of the interviews mentioned that these factors helped facilitate and engage older adults in groups and services. It was also mentioned that active relationships built on honesty, trust, and consistency also helped support engagement and lasting behavioural changes. Furthermore, these relationships fostered a sense of belonging and motivated individuals to continue to participate. A club member stated the following:

Oh, I’d have to say that it is the communication, and… relationships… Then it’s the interest in what we’re doing. It genuinely is friendship and having someone you can… talk to.

These strong relationships were also important for service providers and helped build rapport and identify community needs. Seven organizations mentioned using local knowledge of the community, via these relationships, helped identify the most appropriate resources to offer to individuals and established social networks to provide “safety nets” for individuals identified as needing additional support. One organization explained the significance of these relationships as follows:

If they’re [the community members] looking for advice, or they want somebody to point them in the right direction… they’ll come in and ask for [organization’s name] workers just because they’re reaching that community.

Organizations also mentioned that when services lacked strong relationships or local knowledge, there was a lack of awareness of the community’s needs and a limited understanding of the underlying complex issues that affect the town.

#### 3.3.2. Empowerment

During interviews, facilitators mentioned a few important factors for motivating group members, empowering them, and appealing to their values and identity. In the organizational interviews, community champions or individuals with strong connections to the community were often employed to motivate or empower participation in activities. It was mentioned that the type of individual who qualified as a “champion” varied depending on demographic, but the “champions” tended to be individuals who were pillars of the community. These individuals often empowered communities to learn new skills, participate in groups, or seek services, leading to increased engagement. Sometimes these individuals were also mentioned as the most appropriate resource to connect a person with to change a person’s attitude, behaviour, and engagement:

We identify within our community, our champions… that we can train. We say, ‘You have that greater confidence, who may just be able to go that step further [than the service]’.

Other sources of empowerment to participate stemmed from shared interests, values, and common identities, or motivation in the activity. As mentioned by one competitive individual (m, 60–69, NL), his reasoning for continuing group sports was “getting the locals together and beating them”. Drawing on shared interests of identity proved to be important in other groups as well, especially religious or volunteer-related organizations. As explained by one participant (f, 80–89, NL), “If they don’t worship god, what do we have in common?” Similarly, the values and beliefs of individuals in the governance of formal clubs and organizations influenced individuals’ attendance. Some participants (f, 60–69, NL) (m, 70–79, NL) noted that clubs run by “good people” motivated them to attend, but that “over-boisterous” (f, 90–99, NL) and “controlling” (m, 60–69, NL) people could make a group less appealing. These views highlight the significance of social dynamics in a club.

#### 3.3.3. Disability and Adjusting for Functionality

The interviews also revealed a diverse spectrum of abilities, interests, and participation levels among older adults in the rural town, which impacted participation. Participants frequently noted physical limitations as barriers to social engagement, expressing frustration at their inability to partake in desired activities: “I want to play [sport] again… maybe I’m too old now” (m, 80–89, L). Organizational leaders echoed this sentiment, with 12 out of the interviewed organizations acknowledging limited access for those with health or mobility issues. One leader remarked, “People are excluded if [they]’ve got a disability. I don’t think we’re… accessible”. Accessibility concerns were compounded by limited parking near cafes, lack of ramps, and uneven pathways in the town, all of which restricted social participation for those with mobility constraints.

Alternatively, a few participants with low loneliness scores described adapting their roles within organizations to accommodate their functional limitations: “[I] told them I don’t want to do [role] anymore… because I’m 81” (m, 80–89, NL). These individuals prioritized social connections and maintained their identity through modified participation.

Age and functionality match emerged as a recurrent theme, highlighting the importance of the fit of the activity to the individual participant. When asked about an activity that would be appealing to her, one participant (f, 90–99, NL) with significant disability noted “Somewhere I can use my mind… I’m 91 years [old]… Don’t want to be physical”. Others sought more active engagements, dismissing conventional senior activities with humour: “Oh good god no, that’s only for old farts” (m, 80–89, NL). This sentiment was echoed by a retiree who left a solitary group, emphasizing a preference for active participation:

I did… start going to [group]… But when I was sitting one day between two ladies with dementia, I just couldn’t do it any longer… it’s just not stimulating… I like activity… I’m a doer, not a sitter.(f, 60–69, L)

## 4. Discussion

Fostering healthy aging is a global public health priority, particularly as the aging population increases [17,21]. Understanding the barriers to social participation, a key predictor of quality of life is necessary to support healthy aging [17,22]. However, despite the increasing public health interest in healthy aging, research on social participation among older adults in rural areas remains limited [17]. Existing initiatives predominantly target urban and suburban populations, leaving scarce rural research and policy [17]. In addition, rural areas differ from urban areas in terms of cultural diversity, lack of access to services, and informal information networks, which complicate the transferability of interventions developed in urban settings [23].

The present study corroborates findings that older adults’ psychosocial needs are complex and varied [24]. Evidence from Denmark and rural Canada supports the diversity of older adults’ needs, and how incorporating the needs of many different kinds of people is crucial to promoting healthy aging through social participation interventions [17,24]. Even those living in the same rural town expressed a wide variety of needs, according to the Canadian study, which is consistent with the present study’s findings [17]. This diversity displays the need for flexibility and integral interventions that incorporate multiple components to create long-lasting, sustainable initiatives [21,25]. In addition, a systematic review of community-based participatory research for active aging highlights that providing older people with autonomy to shape the intervention bolsters inclusivity [21]. The present study supports this, as many organizations cited in the results section allowed older adults to switch their roles in the club and maintain their social networks. Moreover, the above findings reinforce the necessity of incorporating diverse perspectives and accommodating varying preferences among older adults, reflecting a need for flexibility in intervention design.

The complexity of older adults’ psychosocial needs emphasizes the importance of community-based interventions. It is known that interventions involving environmental components are more effective in promoting participation than those focusing on individual barriers or characteristics [24]. Traditional approaches targeting individual behaviours may overlook broader community dynamics crucial for effective social participation [21]. In addition, a finding from health promotion research shows that older adults rely on their networks in health decisions (families and peers), highlighting the importance of community involvement to support older people in social participation and engagement [25].

Community involvement is an important element in developing context-appropriate interventions, ensuring local priorities and cultural barriers are addressed, and fostering community buy-in and ownership [26]. Neglecting community dynamics and local knowledge can lead to ineffective interventions imposed from outside, perpetuating a cycle of unsuccessful programs [26]. This was a prominent theme throughout organizational interviews. As further noted by organizations with high older adult participation, the inclusion of a diverse range of older adults or community champions in program design and implementation helped improve engagement, reduce cultural barriers, and develop initiatives that matched the target groups’ functionality, wants, and needs. These improvements, in turn, allowed for the sustainability of social programs and organizations. Representatives of other programs that were not well established in the community, but had successful attendance, mentioned collaborating with programs that had strong ties to the community, which was noted as another effective approach to developing relationships and local knowledge of the community needs. Considering community-specific dynamics and the adaptability of programs to local contexts is also supported in the literature, emphasizing the importance of this result in developing programs in rural communities as well as other geographic environments [26].

Understanding historical and cultural power dynamics and contexts is also crucial to developing culturally sensitive and well-engaged interventions [21,26]. In the context of this rural community, historically marginalized populations had been subjected to research and “help” by outside organizations. These groups were reported to have left the town without providing feedback to the community, and in multiple cases, this led to harm to the older Aboriginal community as they secluded themselves from services and were wary of new organizations, interventions, or people trying to improve social participation. Similar results have been found in other studies, noting that generational shame and stigma have led many elders to forgo care or seek support from trusted members of their community [27,28,29,30]. This further emphasizes the importance of including community representatives in programs to speak to, encourage, and overcome cultural challenges regarding the participation of individuals with minority identities. Shame and stigma were also mentioned by participants in this study, confirming findings from the literature that these impede older adults from asking for help; these factors need to be considered as a potential barrier to social participation and healthy aging [31,32,33].

In addition, diversity of community input would be especially important to this community, as previous studies show that able-bodied, extroverted older adults are the most likely to participate in co-design community activities and increase the sustainability of programs [21]. This is reflected in comments from organizations about how the same few people lead certain organizations, and these same people are key in the community. However, other voices and people are not heard or taken into consideration. Formal efforts with diversified recruitment methods are one recommendation for assessing the nature and extent of the issue of social isolation and loneliness in older people, as the current system of informal information sharing through word-of-mouth leaves out those older adults with a small social network. This style of informal communication in small, rural environments was noted as quite prominent in interviews. In addition, rural communities, in this study and other reports, experience volunteer burnout and although increased responsibility is shown in the literature to enhance participation, this may not work in all cases [34].

### Strengths and Limitations

The strengths of this study lie in the use of in-depth interviews to gain community perspectives on social participation in older adults and triangulations of individual older residents’ perspectives and those of organizations that provide services to assist them. While the interviews did provide meaningful, in-depth insights, metrics were not measured to support interview findings on social participation [10]. Additionally, due to the limited number of interviews and lack of many interviews with Aboriginal peoples, the results from this study are not representative of the entire community. Despite this, based on the aims of this study, the qualitative methodology, and the size of the town’s population, the findings of this study did identify many of the facilitators and barriers of social participation in the subject rural area. In addition, later interviews provided insights into already existing themes, which suggests that most of the barriers and facilitators had previously been identified. Furthermore, the findings from this study align with other studies in rural Australia and other rural communities in similar socioeconomic contexts, suggesting that recommendations and results could be generalizable to other similar environments [27,35]. Further research should be conducted to gain more diverse perspectives on the barriers and facilitators to social participation to better understand methods to increase engagement. Additionally, the individual interview sample lacked diversity, with only one Aboriginal participant due to location constraints and local research priorities. Future studies might recruit diverse samples from Aboriginal medical clinics to develop a more comprehensive view of rural communities in Australia.

## 5. Conclusions

This study underscores the need for community involvement and culturally sensitive interventions that utilize local assets to empower a diverse range of older adults to socially participate in their rural town. By embracing a community-level approach, integrating local knowledge, and including diverse perspectives, communities might improve the social participation, and quality of life, of older adults in rural communities.

These insights into social participation in a rural environment are important due to participation implications in healthy aging and older adults’ well-being. This study provides considerations and potential recommendations to address barriers to engagement and healthy aging initiatives, which could be used to inform future initiatives as well as existing programs, particularly in rural environments. Findings from this study also add to the growing body of research on aging and support interventions focusing on adjusted functionality and community-led design. We recommend that future research and interventions consider these barriers and facilitators as well as other community-specific challenges when testing future interventions.

## Figures and Tables

**Table 1 ijerph-21-00886-t001:** Sex and age of the individual questionnaire sample.

Age	Female *n* (% of Total)	Male *n* (% of Total)
50–59	0 (0)	2 (9)
60–69	2 (9)	2 (9)
70–79	1 (4)	5 (22)
80–89	3 ^1^ (13)	7 (30)
90–99	1 (4)	0 (0)

^1^ Includes one Aboriginal participant.

**Table 2 ijerph-21-00886-t002:** Select sociodemographic characteristics by loneliness.

Demographic Characteristic	Loneliness	
	Lonely *n* (%)	Not Lonely *n* (%)
Gender		
Female	3 (43)	4 (57)
Male	6 (38)	10 (63)
Age		
50–69	2 (33)	4 (67)
70–99	7 (41)	10 (59)
Health *		
Poor/fair	6 (75)	2 (25)
Good/very good	2 (15)	11 (85)
Marital Status		
Married/de facto	4 (29)	10 (71)
Widowed/divorced	4 (50)	4 (50)
Other	1 (100)	0 (0)
Cohabitation ^a^	4 (33)	8 (66)

^a^ Reflects the number and percentage of participants answering “yes” to this question. * Two participants declined to answer.

**Table 3 ijerph-21-00886-t003:** Organizational characteristics of the organizational interview sample.

Organizational Characteristics	Number of Organizations *n* (%)
Function	
Governmental	3 (15)
Social service	6 (32)
Social group/club	10 (53)
Structure	
Branch	15 (79)
Independent	4 (21)

**Table 4 ijerph-21-00886-t004:** Representative characteristics of the organizational interview sample.

Representative Characteristics	Number of Participants *n* (%)
Gender	
Female	23 (85)
Male	4 (15)
Race/ethnicity	
Aboriginal	3 (11)
Non-Aboriginal	23 (85)
Asian	1 (4)
Position	
Leadership	15 (56)
Non-leadership	12 (44)

## Data Availability

The data from this study are available upon request from the corresponding author. The data are not publicly available due to privacy restrictions.

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
