# Peer review of "Social Participation and Loneliness in Older Adults in a Rural Australian Context: Individual and Organizational Perspectives"

_ijerph, 2024, doi:10.3390/ijerph21070886_

Round 1
Reviewer 1 Report
Comments and Suggestions for Authors
This is a well written and interesting manuscript on social engagement among rural-dwelling older adults - an research area of increasing importance given trends in population ageing and internal migration. I have the following suggestions for changes that I think would strengthen the report:
General
The main focus of the manuscript is social participation – loneliness is also in the title but gets little coverage. I think that either more needs to be discussed relating to loneliness throughout the paper or loneliness should be dropped from the title.
Introduction
In the first or second paragraph, adding a little more information on why ageing populations are seen as a problem would help provide context.
Methods
Please could you add inclusion/exclusion criteria for your sample (e.g., what was the minimum age, were people with, say, cognitive impairment or who could not speak English included?). It would also be helpful to give an idea of how many older adults were seen at the medical practice and approached to take part in the study (or perhaps add this to the Results). Information about the different ethnic groups present in the town would be useful (particularly proportion of Aboriginal and White). Were any measures taken to increase diversity/inclusion in your sample?
The consent process is described for older adult participants but not for staff participants - please could this be added. Please also state how many organisations were identified as relevant/approached. It would also be helpful to understand how representatives from these organisations were selected.
Please could you explain why transcripts summaries were returned to staff for checking but not to older adults? Also, were any transcript summaries changed based on participant’s checks?
The first sentence in the third paragraph in subsection 2.4 is repeated.
Please provide the interview schedules as appendices/supplementary materials.
To help readers assess how the researchers’ experiences and thoughts might have shaped the research, it would be helpful to provide a reflexivity and positionality statement for the researchers conducting the interviews and analysis.
Results
The third paragraph is repeated on either side of Table 1.
It would be helpful to get an idea of how many older people the organisations/social groups served and how long they had been running, if this information is available.
You used the de Jong Gierveld loneliness scale in your interviews with older adults – please could you provide the results from this?
It wasn’t clear to me what the difference between Sustainability and Sustainable Programmes was. Could this be clarified or the two sub-themes merged?
In the final paragraph of 3.2.2, there seems to be an opening quotation mark missing
Discussion
Sustainability was a key theme but is not mentioned in the Discussion – could more be added on this, perhaps tying it in with community involvement.
If mentioning data saturation, please indicate at what point you feel this was reached and how it was determined. However, I would encourage the authors to avoid discussing saturation and instead comment on the adequacy of their data in relation to their aims (see Vasileiou et al 2018 https://doi.org/10.1186/s12874-018-0594-7).
Author Response
Thank you very much for your time and effort spent on this feedback. We are grateful for your detailed feedback. Below are our responses to your comments.
1.This is a well written and interesting manuscript on social engagement among rural-dwelling older adults - an research area of increasing importance given trends in population ageing and internal migration. I have the following suggestions for changes that I think would strengthen the report:
Thank you!
- The main focus of the manuscript is social participation – loneliness is also in the title but gets little coverage. I think that either more needs to be discussed relating to loneliness throughout the paper or loneliness should be dropped from the title.
Thank you for this point. We have included some information about loneliness in the introductions and methods, statistical tables in the results, and labels for participant quotes based on their loneliness status to signify how loneliness fits with the main theme of social participation. “Importantly, it is estimated that 34% of Australian adults feel lonely [6]. Social participation can reduce this loneliness as well as social isolation, which are linked to health and social challenges [7-9].”
- In the first or second paragraph, adding a little more information on why ageing populations are seen as a problem would help provide context.
As suggested, we have included some information from a source about the health system burden posed by aging populations. “The population change places pressure on social and health services to sustain an increased demand for medical and aged care made by older adults in Australia [1,2].”
- Please could you add inclusion/exclusion criteria for your sample (e.g., what was the minimum age, were people with, say, cognitive impairment or who could not speak English included?).
We agree with this suggestion. Accordingly, we added a section for inclusion/exclusion criteria: “Each participant, aged 50 and above, was referred by health personnel to speak with researchers to ascertain their interest in participation. This clinic sample was based on patient availability. The discretion of medical providers was used to determine which participants were cognitively able to participate. In addition, any participants who were not English-speaking were excluded.”
- It would also be helpful to give an idea of how many older adults were seen at the medical practice and approached to take part in the study (or perhaps add this to the Results).
Thank you for the suggestion. We wish we could include this but it is unknown to the authors how many adults were approached, as doctors/nurses at the medical center were asked to recruit for the study but did not collect that information.
- Information about the different ethnic groups present in the town would be useful (particularly proportion of Aboriginal and White).
We agree with this suggestion! We had added this in the Methods: “About 16% of the population is identified as Aboriginal, 67% as White, and 17% as unknown [18].”
- Were any measures taken to increase diversity/inclusion in your sample?
The town studied has two medical centers, Aboriginal Medical Service (AMS) and Carnarvon Medical Centre (CMC). We asked to recruit through both, however the AMS denied permission. Thus, we only recruited through the CMC and therefore had few Aboriginal respondents.
- The consent process is described for older adult participants but not for staff participants - please could this be added. Please also state how many organisations were identified as relevant/approached. It would also be helpful to understand how representatives from these organisations were selected.
Thank you. The consent process for organizations, number of organizations contacted, and selection processes for representatives are now included: “Representatives were chosen based on their leadership status within the organization and availability.” “Prior to the interviews, all participants were informed of the study purpose, given study information forms and consent forms that explained the interview process, and advised that the interview would be recorded. Participants who agreed to participate signed written consent forms.” “Nineteen of the thirty-nine organizations that were identified and contacted agreed to participate.”
- Please could you explain why transcripts summaries were returned to staff for checking but not to older adults? Also, were any transcript summaries changed based on participant’s checks?
Yes. Summaries were provided to organizations because many interviewers were not representing themselves but the organization’s larger mission and initiatives. In order to insure they were not misrepresented we created summaries. We did not collect contact information from individual participants (for privacy) and therefore did not provide summaries. Email also would have been a challenge for many of the older adult participants who were not accustomed to computers.
- The first sentence in the third paragraph in subsection 2.4 is repeated.
Thank you for this comment. It is now removed.
- Please provide the interview schedules as appendices/supplementary materials.
Thank you for catching this. They are now included as supplementary materials.
- To help readers assess how the researchers’ experiences and thoughts might have shaped the research, it would be helpful to provide a reflexivity and positionality statement for the researchers conducting the interviews and analysis.
We appreciate your suggestion and have included a reflexivity and positionality statement. “We acknowledge our standpoint as educated, White Americans and Australians. In terms of rurality, while the researchers lived in the community during the project, we remained outsiders to this remote area. In addition, the primary authors have age-related privilege, and do not have lived experience in aging.”
- The third paragraph is repeated on either side of Table 1.
Thank you. It is removed.
- It would be helpful to get an idea of how many older people the organisations/social groups served and how long they had been running, if this information is available.
This would be helpful, however, many organizations had targets for participation but did not meet these goals or had inconsistent numbers. Additionally, some organizations canceled programs or were inactive periodically because of the transient nature of the town or lack of funding. Thus, this data is largely unknown.
- You used the de Jong Gierveld loneliness scale in your interviews with older adults – please could you provide the results from this?
Yes, the results are provided in a table and in the results section. Additionally, participant quotes are labeled with their loneliness results.
- It wasn’t clear to me what the difference between Sustainability and Sustainable Programmes was. Could this be clarified or the two sub-themes merged?
Thank you. The Sustainability section included changes within the town (including population movement) and lack of volunteers for general events. The subsection of sustainable programs included aspects that pertained to the sustainability of organizations and programs only.
- In the final paragraph of 3.2.2, there seems to be an opening quotation mark missing
Thank you, it is fixed.
- Sustainability was a key theme but is not mentioned in the Discussion – could more be added on this, perhaps tying it in with community involvement.
We agree with this suggestion, and have included sentences within the paper: “These improvements in turn allowed for sustainability of social programs and organizations.” “This diversity displays the need for flexibility and integral interventions that incorporate multiple components in order to create long lasting, sustainable initiatives [25,21]”.
- If mentioning data saturation, please indicate at what point you feel this was reached and how it was determined. However, I would encourage the authors to avoid discussing saturation and instead comment on the adequacy of their data in relation to their aims (see Vasileiou et al 2018 https://doi.org/10.1186/s12874-018-0594-7).
Thank you. We have reviewed the article you included and, based on the information, included this in our “Strengths and Limitations” section: “Despite this, based on the aims of the study, qualitative methodology, and size of the town’s population, the findings of this study did identify many of the facilitators and barriers of social participation in this rural area. In addition, later interviews provided insights into already existing themes, which suggests that most of the barriers and facilitators had previously been identified.”
Reviewer 2 Report
Comments and Suggestions for Authors
I enjoyed reading your article, and appreciated you tackling an under-researched topic. Lack of sufficient volunteers, third spaces and infrastructure in remote and rural regions is something that attention needs to be drawn to, in order to assist with the sustainability of social activities, and your article helps with this. The importance of informal networks and local community comes out in many parts of your article, yet these do not seem to be adequately supported. Hopefully, your article will assist with future policy choices.
The sample size is quite small, biased towards males and does not have enough First Nations people. Perhaps the authors could suggest how a more representative sample could be achieved? Why are males the main respondents?
The authors suggest that making 'third spaces' available for social interaction is an important policy response (p7), a comment I agree with, Perhaps the authors could elaborate on how this can be achieved (funding? organisation?).
On cultural barriers, the authors highlight "rural stoicism, town culture, 263 local versus outsider power dynamics, Aboriginal versus non-Aboriginal tensions, 264 and intergenerational trauma and shame." (p8) Their discussion of these (difficult) issues is welcome. However, I would have welcomed a bit more discussion about what can be done to alleviate these problems.
The Canadian study cited on p12 has some interesting results. Would it be possible to replicate such a study in Australia? Does Canada have better data, and if so what can be done to improve Australian data?
Author Response
We appreciate your insight that you dedicated to providing feedback on our manuscript. We have incorporated most of the suggestions made.
- I enjoyed reading your article, and appreciated you tackling an under-researched topic. Lack of sufficient volunteers, third spaces and infrastructure in remote and rural regions is something that attention needs to be drawn to, in order to assist with the sustainability of social activities, and your article helps with this. The importance of informal networks and local community comes out in many parts of your article, yet these do not seem to be adequately supported. Hopefully, your article will assist with future policy choices.
Thank you, we appreciate your insight!
- The sample size is quite small, biased towards males and does not have enough First Nations people. Perhaps the authors could suggest how a more representative sample could be achieved?
We agree with this comment. We were not able to work with the Aboriginal medical clinics but agree that future studies should try for a more diverse sample. “Future studies might recruit diverse samples from Aboriginal medical clinics to develop a more comprehensive view of rural communities in Australia.”
- Why are males the main respondents?
The medical personnel at the medical centre were asked to recruit patients for the study. It is unclear whether more men agreed to participate, or if there were more male patients during that week. We agree that it is quite unusual, but the men provided an interesting rural stoic perspective that added to themes.
- The authors suggest that making third spaces available for social interaction is an important policy response (p7), a comment I agree with, Perhaps the authors could elaborate on how this can be achieved (funding? organisation?).
This is a great insight. We included a brief detail to add to our information about third spaces: “The desire for increased availability and accessibility of such spaces reflects a broader need for infrastructure, including parks, gardens, and local businesses, that support social gatherings and cohesion in rural settings.”
- On cultural barriers, the authors highlight rural stoicism, town culture, local versus outsider power dynamics, Aboriginal versus non-Aboriginal tensions, and intergenerational trauma and shame. Their discussion of these (difficult) issues is welcome. However, I would have welcomed a bit more discussion about what can be done to alleviate these problems.
Thank you! Cultural barriers can be addressed by including more community representation in planning and implementing social programing and other services. The discussion includes comments regarding ways to increase diversity and community participation. Including comments such as “This further emphasizes the importance of including community representatives in programs to speak to, encourage, and overcome cultural challenges in the participation of individuals with shared identities.”
- The Canadian study cited on p12 has some interesting results. Would it be possible to replicate such a study in Australia? Does Canada have better data, and if so what can be done to improve Australian data?
This is an interesting idea. The Canadian study included focus groups and interviews with individuals and organizations, similar to our own study. More funding would be helpful to support new studies in rural areas, however as we encountered, even existing services lack sufficient funding.